# Core-Shell Carbon Nanofibers@Ni(OH)_2_/NiO Composites for High-Performance Asymmetric Supercapacitors

**DOI:** 10.3390/ma15238377

**Published:** 2022-11-24

**Authors:** Peizhi Fan, Lan Xu

**Affiliations:** National Engineering Laboratory for Modern Silk, College of Textile and Engineering, Soochow University, Suzhou 215123, China

**Keywords:** Ni(OH)_2_, NiO, carbon fiber, coexisted

## Abstract

The application of transition metal oxides/hydroxides in energy storage has long been studied by researchers. In this paper, the core-shell CNFs@Ni(OH)_2_/NiO composite electrodes were prepared by calcining carbon nanofibers (CNFs) coated with Ni(OH)_2_ under an N_2_ atmosphere, in which NiO was generated by the thermal decomposition of Ni(OH)_2_. After low-temperature carbonization at 200 °C, 250 °C and 300 °C for 1 h, Ni(OH)_2_ or/and NiO existed on the surface of CNFs to form the core-shell composite CNFs@Ni(OH)_2_/NiO-X (X = 200, 250, 300), in which CNFs@Ni(OH)_2_/NiO-250 had the optimal electrochemical properties due to the coexistence of Ni(OH)_2_ and NiO. Its specific capacitance could reach 695 F g^−1^ at 1 A g^−1^, and it still had 74% capacitance retention and 88% coulomb efficiency after 2000 cycles at 5 A g^−1^. Additionally, the asymmetric supercapacitor (ASC) assembled from CNFs@Ni(OH)_2_/NiO-250 had excellent energy storage performance with a maximum power density of 4000 W kg^−1^ and a maximum functional capacity density of 16.56 Wh kg^−1^.

## 1. Introduction

Developing a reliable power storage system is one of the important ways to solve the pollution of traditional fuel energy. As a new generation of energy storage systems, supercapacitors have gained much concern in the field of energy storage due to their fast charging/discharging rate, excellent cycling stability and high power density. Typically, supercapacitors can be divided into double-layer capacitors and pseudocapacitor capacitors. The double-layer capacitors mainly store energy by generating an accumulation of electrostatic charges at the interface between electrode and electrolyte, and their electrode materials are mainly carbon materials, such as activated carbon (AC), carbon nanotubes and carbon nanofibers (CNFs) [1,2,3]. The pseudocapacitors achieve the electrochemical storage of electrical energy through Faraday redox reactions, and their electrode materials are mainly transition metal oxides and conducting polymers [4,5,6]. However, consisting of two double-layer capacitive electrode materials or two pseudocapacitive electrodes, they usually have a low energy density (5–10 Wh kg^−1^) [7]. In order to improve the energy density of supercapacitors, asymmetric supercapacitors with a double-layer capacitive electrode and a pseudocapacitive electrode have received increasing attention. The combination of the two electrodes produces a wider voltage window, which makes the energy density of asymmetric supercapacitors often greater than that of symmetric supercapacitors [8].

Among a large number of electrode materials, Ni(OH)_2_ has a high theoretical specific capacitance and layered structure, while NiO has a higher specific capacitance, low price and easy synthesis [9]. Therefore, Ni(OH)_2_ and NiO have received more and more attention and are often used as electrodes in asymmetric supercapacitors due to their excellent electrochemical properties [10]. However, due to their poor crystallinity, electrical conductivity and redox reversibility, their real capacitance values in practical applications are far lower than the ideal values [11]. An effective way to cope with this problem is to prepare them into nanostructured materials, such as nanoflowers [12] and nanosheets [13], which can effectively increase the specific capacitance of the material. Another approach is to combine Ni(OH)_2_ and NiO or compound them with other materials into composite materials, which can effectively improve the performance of a single material. For example, Dai et al. prepared Ni(OH)_2_/NiO/Ni composite nanotube array films by electrodeposition, which showed that the multi-layered composites had high specific capacitance, excellent multiplicative properties and excellent electrochemical cycling stability [14]. Gunasekaran et al. fabricated NiO@Ni(OH)_2_-α-MoO_3_ heterostructured nanocomposites with high specific capacitance, long cycle stability, excellent reversibility and high energy/power density by hydrothermal process [15]. In addition, the composites of Ni(OH)_2_ or NiO with carbon-based materials can enhance their electronic conductivity, increase their specific surface areas and improve their capacitive properties [16], which are excellent candidates for supercapacitor electrode materials. Electrospinning technology is a simple and effective method to fabricate nanofibers [17], and the carbon nanofibers (CNFs) derived from the carbonization of nanofibers have excellent electrical conductivity and good mechanical stability, which are often used as carbon-based electrode materials for supercapacitors [18]. However, Ni(OH)_2_ and NiO co-grown on CNFs have rarely been reported.

In this work, core-shell composites (CNFs@Ni(OH)_2_/NiO-X (X = 200, 250, 300)) were prepared by co-coating CNFs with Ni(OH)_2_ and NiO through electrospinning, high-temperature carbonization, hydrothermal synthesis and low-temperature carbonization in sequence, as shown in Figure 1. It could be found that a layer of Ni(OH)_2_ was grown on the surface of CNFs after the hydrothermal process. Due to the unstable structure of Ni(OH)_2_, the CNFs coated with Ni(OH)_2_ were subsequently subjected to low-temperature carbonization at 200 °C, 250 °C, and 300 °C for 1 h in the N_2_ atmosphere, which could protect the CNFs from depletion and obtain core-shell composites with the coexistence of Ni(OH)_2_ and NiO. Compared with Ni(OH)_2_, the structure of NiO is undoubtedly more stable [11]. Therefore, a core-shell composite with Ni(OH)_2_ and NiO as the shell layer and CNFs as the core layer (CNFs@Ni(OH)_2_/NiO-250) was obtained at 250 °C. The electrochemical properties of CNFs@Ni(OH)_2_/NiO-X were characterized, and it was found that CNFs@Ni(OH)_2_/NiO-250 had the best electrochemical properties because of the coexistence of Ni(OH)_2_ and NiO. Accordingly, the energy storage performance of the asymmetric supercapacitor (ASC) assembled from CNFs@Ni(OH)_2_/NiO-250 was investigated.

## 2. Experimental Section

### 2.1. Preparation of CNFs@Ni(OH)_2_/NiO

As shown in Figure 1, 4 g polyacrylonitrile (PAN) was weighed and dissolved in 38 g N,N-Dimethylformamide (DMF) and stirred for 24 h, then 0.72 g Ni(CH_3_COO)_2_ was added and stirred for 6 h and sonicated for 0.5 h to obtain the spinning precursor solution for electrospinning. The spinning voltage was 16 kv, the flow rate was 1 mL h^−1^, and the receiving distance was 18 cm, thus obtaining the nanofiber membranes (NFMs) containing Ni^2+^.

The electrospun NFMs were firstly pre-oxidized in a muffle furnace in an air atmosphere at a heating rate of 2 °C min^−1^ to 250 °C and held for 2 h. Then, the pre-oxidized NFMs were carbonized in a tube furnace at a heating rate of 5 °C min^−1^ to 800 °C for 2 h under an N_2_ environment. Next, the carbonized NFMs were immersed in a hydrothermal growth solution consisting of 1.16 g Ni(NO_3_)_2_, 2.4 g urea and 80 mL deionized water for 1 h, and then the hydrothermal growth solution and NFMs were transferred to the polytetrafluoroethylene (PTFE) liner together for hydrothermal reaction with 120 °C reaction temperature and 6 h reaction time. The NFMs obtained after the hydrothermal reaction were repeatedly washed with deionized water and ethanol, and dried at 60 °C. Finally, the above NFMs were placed in a tube furnace again and warmed up to 200, 250 and 300 °C, respectively, in an N_2_ atmosphere at a rate of 5 °C min^−1^ and held for 1 h. The obtained materials were noted as CNFs@Ni(OH)_2_/NiO-X (X = 200, 250, 300).

### 2.2. Preparation of CNFs@Ni(OH)_2_/NiO-X Electrodes and ASC Assembly

Firstly, a piece of nickel foam was placed in hydrochloric acid and ethanol for ultrasonic washing, respectively, and then cut into 1 × 2 cm^2^ rectangular strips and set aside. Then, CNFs@Ni(OH)_2_/NiO-X powders, acetylene black (conductive agent) and PTFE emulsion (binder) were weighed separately in the ratio of 8:1:1 and mixed well, and then a small amount of ethanol solution was dropped to prepare a uniform paste-like slurry. Afterwards, the prepared paste slurry was evenly applied to a 1×1 cm^2^ area of the nickel foam with a clean spatula. After drying the coated nickel foam in an oven for 12 h, it was placed under a pressure of 10 MPa for about 10 s to prepare CNFS@Ni(OH)_2_/NiO-X electrodes. The final mass of the active substances coated on the electrode was 4 mg. The AC electrode was prepared in the same way, and the mass of the active substances on the AC electrode was calculated according to Equation (1).
(1)$$\frac{m_{+}}{m_{-}}=\frac{C_{-}*\Delta V_{-}}{C_{+}*\Delta V_{+}}$$
where + and − represent the positive and negative electrodes, respectively, *m* (g) represents the mass of active substances, *C* (F g^−1^) represents the specific capacitance, and Δ*V* (V) represents the range of voltage window.

The CNFS@Ni(OH)_2_/NiO-250 electrode was used as the positive electrode of an asymmetric supercapacitor (ASC), the AC electrode was used as the negative electrode of the ASC, and 3 M KOH solution was used as the electrolyte, which was assembled in an electrolytic cell to form CNFs@Ni(OH)_2_/NiO-250//AC ASC.

### 2.3. Electrochemical Performance Characterization

The electrochemical performance of electrodes was tested using a CHI660E electrochemical workstation for a three-electrode system. The platinum sheet electrode was used as the auxiliary electrode, the HgO/Hg electrode was used as the reference electrode, the CNFs@Ni(OH)_2_/NiO-X electrode was used as the working electrode, and 3 M KOH solution was used as the electrolyte.

The specific capacitance of the single electrode is calculated according to the GCD curve, which is shown in Equation (2).
(2)$$C=\frac{I*\Delta t}{m*\Delta V}$$
where *I* (A) is the current, and ∆*t* (s) is the discharge time.

The energy density and power density of ASC are calculated using Equations (3) and (4).
(3)$$E=\frac{1}{2}*\frac{1}{3.6}*C*\Delta V^{2}$$
(4)$$P=3600*\frac{E}{\Delta t}$$
where *E* (Wh kg^−1^) is the energy density, and *P* (W kg^−1^) is the power density.

## 3. Results

### 3.1. Structure and Morphology Analysis

The physical phase composition of CNFs@Ni(OH)_2_/NiO-X was determined by X-ray diffraction (XRD) tests, and their XRD patterns were shown in Figure 2a. The characteristic peaks at 11.3°, 33.6°, 35.1° and 59.9° in the curves corresponded to the (001), (110), (111) and (300) crystal planes of Ni(OH)_2_, respectively (JCPDS: 22-0444). The characteristic peaks at 37.1°, 43.1° and 62.6° in the curves corresponded to the (111), (200) and (220) crystal planes of NiO, respectively (JCPDS: 89-7130). The characteristic peak at 24° in the curves was the (002) crystal plane of CNFs (JCPDS: 20-0018). The XRD patterns of CNFs@Ni(OH)_2_/NiO-X indicated that the characteristic peaks of CNFs@Ni(OH)_2_/NiO-200 were mainly in the Ni(OH)_2_ phase. With the increase in temperature, CNFs@Ni(OH)_2_/NiO-250 showed the co-existence of Ni(OH)_2_ and NiO characteristic peaks. With the further increase in temperature, the characteristic peaks of CNFs@Ni(OH)_2_/NiO-300 were mainly in the NiO phase. Therefore, it could be considered that as the calcination temperature increased, CNFs@Ni(OH)_2_/NiO-X surface-derived materials gradually overstepped from Ni(OH)_2_ to NiO because NiO was generated by the decomposition of Ni(OH)_2_ at high temperature. In the 250°C environment, Ni(OH)_2_ and NiO were in a co-existing state.

The Raman spectra of CNFs@Ni(OH)_2_/NiO-X (Figure 2b) displayed that their curves contained two distinctive characteristic peaks located at about 1340 cm^−1^ and about 1575 cm^−1^, respectively, corresponding to the D-band and G-band of the C-atom crystal. The I_D_/I_G_ of CNFs@Ni(OH)_2_/NiO-200 was 0.54, that of CNFs@Ni(OH)_2_/NiO-250 was 0.84, and that of CNFs@Ni(OH)_2_/NiO-300 was 0.44. This indicated that CNFs@Ni(OH)_2_/NiO-250 had the highest degree of disorder, and the higher degree of disorder was beneficial to promote the rapid movement of electrons between its graphite layers and improve the electrochemical performance of the electrode material.

The morphology of CNFs@Ni(OH)_2_/NiO-X was characterized by scanning electron microscopy (SEM). As shown in Figure 3a and XRD analysis results, it was found that many uniformly distributed Ni(OH)_2_ nanosheets were grown on the surface of CNFs@Ni(OH)_2_/NiO-200. The number of Ni(OH)_2_ nanosheets on the surface of CNFs@Ni(OH)2/NiO-250 decreased, but many nanoparticles appeared, indicating the co-existence of Ni(OH)_2_ nanosheets and NiO nanoparticles (Figure 3b). Additionally, the surface of CNFs@Ni(OH)_2_/NiO-300 was significantly free of Ni(OH)_2_ nanosheets, which had been transformed into NiO nanoparticles (Figure 3c).

The elemental contents of CNFs@Ni(OH)_2_/NiO-X obtained by energy dispersive X-ray spectrometry (EDS) were shown in Table 1 and Appendix A. It could be seen that the composites mainly contained C, N, O and Ni. As the calcination temperature increased, the content ratio of O atoms to Ni atoms in the composites gradually became smaller due to the conversion of some Ni(OH)_2_ into NiO.

The morphology and structure of CNFs@Ni(OH)_2_/NiO-250 were further explored by high-resolution transmission electron microscopy (TEM), as shown in Figure 4. Figure 4a,b clearly showed that CNFs@Ni(OH)_2_/NiO-250 had a distinct core-shell structure, and its surface was rough with a large number of gaps, which were beneficial to increase its specific surface area, improving its contact area with the electrolyte, and enhance its electrochemical properties. Figure 4c–e exhibited that the lattice stripe spacing of d = 0.254 nm corresponded to the (111) crystal plane of Ni(OH)_2_, and the lattice stripe spacing of d = 0.209 nm and d = 0.244 nm corresponded to the (200) and (111) crystal planes of NiO, respectively. This further supported the coexistence of Ni(OH)_2_ and NiO on CNFs@Ni(OH)_2_/NiO-250.

The chemical compositions and valences of CNFs@Ni(OH)_2_/NiO-250 were investigated by X-ray photoelectron spectroscopy (XPS), as displayed in Figure 5 and Appendix A. It could be found that CNFs@Ni(OH)_2_/NiO-250 contained C, N, O and Ni (Figure 5a). In its Ni2p XPS spectrum (Figure 5b), the binding energy of 854.5 ev versus 872.3 ev was attributed to the characteristic peak at Ni2p 3/2 versus Ni2p 1/2, which contributed to the redox reactions of +3 and +2 states of Ni(OH)_2_ and NiO in the electrochemical performance [19,20]. Compared with the characteristic peak at 855.6 ev for Ni2p 3/2 of pure Ni(OH)_2_, the characteristic peak was shifted to the right due to the partial production of NiO by calcination, while the characteristic peak for Ni2p 3/2 of pure NiO was at 853.7 ev. The characteristic peaks at binding energies of 860.5 ev and 878.3 ev were the satellite peaks of Ni2p 3/2 and Ni2p 1/2, respectively. The O1s XPS spectrum (Figure 5c) could be divided into three characteristic peaks, namely C-O (533.1 ev), C=O (531.5 ev), M-O (530.0 ev), and the C1s XPS spectrum (Appendix A) were also divided into three characteristic peaks, namely C-C (284.7 ev), C-O (286.1 ev) and C=O (288.5 ev).

The specific surface area and pores of CNFs@Ni(OH)_2_/NiO-250 were characterized by N_2_ adsorption-desorption (BET) tests, as illustrated in Figure 5d and Table 2. It could be seen that the pores of CNFs@Ni(OH)_2_/NiO-250 were mainly intermediate pores, with a BET specific surface area of 49.2990 m^2^ g^−1^ and a pore volume of 0.116 cm^3^ g^−1^. A large number of intermediate pores was conducive to the full contact between the composite and the electrolyte, thus promoting rapid ion transport.

### 3.2. Electrochemical Property 

Figure 6 and Appendix A showed the electrochemical properties of CNFs@Ni(OH)_2_/NiO-X. It was clear from the CV curves of CNFs@Ni(OH)_2_/NiO-250 at different sweep rates (Figure 6a) that there were obvious redox peaks, indicating that the redox reaction had a significant contribution to the capacitance of the composite [9,21]. The redox reaction was mainly related to Equations (5) and (6). Moreover, the CV curves of CNFs@Ni(OH)_2_/NiO-X at 5 mV s^−1^ sweep rate (Figure 6b) exhibited that all three composites had distinct redox peaks, and the CV curve area of CNFs@Ni(OH)_2_/NiO-250 was the largest, proving that the specific capacitance of CNFs@Ni(OH)_2_/NiO-250 was the largest.
(5)$$\mathrm{NiO}+{\mathrm{OH}}^{-}↔\mathrm{NiOOH}+e^{-}$$
(6)$$\mathrm{Ni}{\left(\mathrm{OH}\right)}_{2}+{\mathrm{OH}}^{-}↔\mathrm{NiOOH}+H_{2}O+e^{-}$$

The GCD curves of CNFs@Ni(OH)_2_/NiO-250 at different current densities (Figure 6c) had clear charging and discharging plateaus, which were related to the redox reaction. Additionally, the GCD curves of CNFs@Ni(OH)_2_/NiO-X at 1 A g^−1^ (Appendix A) displayed that all three composites had significant charging and discharging plateaus, and CNFs@Ni(OH)_2_/NiO-250 had the longest discharge time, which also further assisted to prove that it had the highest specific capacitance. Figure 6d illustrated the specific capacitances of CNFs@Ni(OH)_2_/NiO-X at different current densities. It could be observed that the specific capacitance of CNFs@Ni(OH)_2_/NiO-250 was highest at the same current density, and its specific capacitance values at 1, 3, 5, 8, 10, and 20 A g^−1^ were 695, 615, 562.5, 480, 400, and 220 F g^−1^, respectively. This indicated that the coexistence of Ni(OH)_2_ and NiO was beneficial to improve the electrochemical properties of materials. Compared with other reported Ni-based electrode materials (Appendix A), CNFs@Ni(OH)_2_/NiO-250 still exhibited good specific capacitance.

The electrochemical impedance (EIS) diagram of CNFs@Ni(OH)_2_/NiO-X was shown in Figure 6e, and the equivalent circuit diagram of Nyquist plots was given in the inset. The intersection of the semicircular curve and the real axis in the high-frequency region is the ohmic resistance (Rs) in the solution [22], and the semicircle diameter is related to the charge transfer resistance (Rct). The approximate straight line in the low-frequency region is caused by the diffusion resistance (Zw) [23,24]. It could be seen from Figure 6e that CNFs@Ni(OH)_2_/NiO-X exhibited small charge transfer resistance in high-frequency regions and nearly vertical lines in low-frequency regions, which indicated their good ion diffusion mobility and rapid electron transport, resulting in their excellent capacitance characteristics [2]. Figure 6f illustrated that CNFs@Ni(OH)_2_/NiO-250 still had 74% capacitance retention and 88% coulomb efficiency after 2000 cycles at 5 A g^−1^, indicating its good cycling stability.

According to the CV curves of CNFs@Ni(OH)_2_/NiO-250 at different sweep rates, its charge storage mechanism was further investigated [25]. The relationship between the capacitance and diffusion-controlled contributions can be determined by Equations (7) and (8) [26].
(7)$$i_{p}=av^{b}$$
(8)$$i=k_{1}v+k_{2}v^{0.5}$$
where *a* and *b* are adjustable parameters, *i* is the peak current (mA), *v* is the scan rate (mV/s), *k*_1_*v* is the capacitance contribution, and *k*_2_*v*^0.5^ is the diffusion control contribution.

In Equation (7), if the b value is close to 0.5, the reaction mechanism is mainly the battery type where ion diffusion occurs. If the b value is close to 1, the reaction mechanism is mainly contributed by the capacitance. As shown in Figure 7a, the b-values of the anode and cathode were 0.549 and 0.529, respectively, which could be inferred that the reaction mechanism of CNFs@Ni(OH)_2_/NiO-250 electrode was a coexisting state of capacitive and battery-type charge storage mechanism. Figure 7b displayed that as the sweep rate increased, the diffusion-controlled contribution gradually decreased, and the capacitive contribution gradually increased. When the sweep rate increased from 2.5 mV s^−1^ to 30 mV s^−1^, the diffusion-controlled contribution decreased from 59.4% to 30%.

To determine a reasonable voltage window for CNFs@Ni(OH)_2_/NiO-250//AC ASC, its CV curve from 0–2 V at 50 mV s^−1^ was tested (Figure 8a). Based on the distribution of the maximum area of the CV curve, a voltage window of 0–1.6 V was finally selected for the ASC device. Figure 8b showed the CV curves of CNFs@Ni(OH)_2_/NiO-250//AC ASC at sweep rates of 5, 10, 20, 30, and 40 mV s^−1^. It could be seen from the curves that the ASC had a stable morphology at a voltage window of 1.6 V. Figure 8c displayed the GCD curves of ASC at 1, 2, 3, 4, and 5 A g^−1^ were basically symmetrical, indicating that ASC has good electrochemical reversibility. According to the GCD curves, the specific capacitances of CNFs@Ni(OH)_2_/NiO-250//AC ASC at 1, 2, 3, 4, 5 A g^−1^ were 46.5, 35, 28.1, 25, and 21.9 F g^−1^, respectively, and its energy density versus power density curve was plotted to evaluate its energy storage performance, as shown in Figure 8d. It could be found that CNFs@Ni(OH)_2_/NiO-250//AC ASC had an energy density of 7.78 Wh kg^−1^ at a maximum power density of 4000 W kg^−1^ and a power density of 800 Wh kg^−1^ at a maximum energy density of 16.56 Wh kg^−1^. The energy density and power density of the CNFs@Ni(OH)_2_/NiO-250//AC ASC device was compared with the data already reported [15,16,27,28,29,30,31,32,33], as shown in Figure 8d.

## 4. Conclusions

In summary, a core-shell composite with the coexistence of NiO and Ni(OH)_2_ (CNFs@Ni(OH)_2_/NiO-250) was designed in this work. NiO was obtained by relying on the thermal decomposition of Ni(OH)_2_ at low temperatures, which could protect the CNFs from loss. The specific capacitance of CNFs@Ni(OH)_2_/NiO-250 could reach 695 F g^−1^ at a current density of 1 A g^−1^, and it still had 74% capacitance retention and 88% coulomb efficiency after 2000 cycles at 5 A g^−1^, showing its good specific capacitance and cycling stability. Additionally, the CNFs@Ni(OH)_2_/NiO-250//AC ASC device also exhibited excellent energy storage performance with a maximum power density of 4000 W kg^−1^ and a maximum functional capacity density of 16.56 Wh kg^−1^.

## Figures and Tables

**Figure 1 materials-15-08377-f001:**
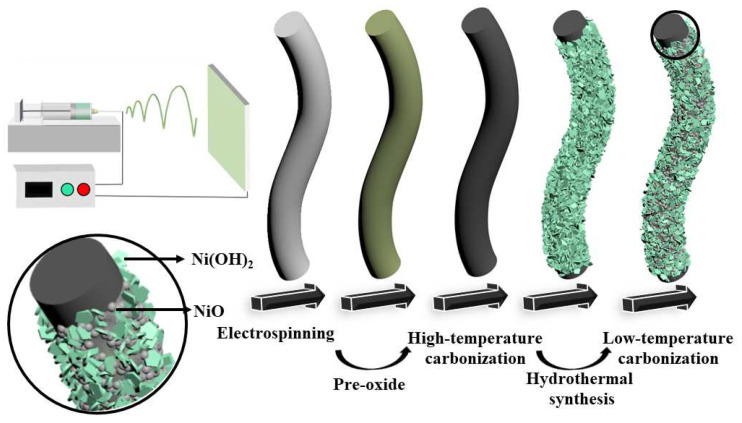
Preparation of CNFs@Ni(OH)_2_/NiO.

**Figure 2 materials-15-08377-f002:**
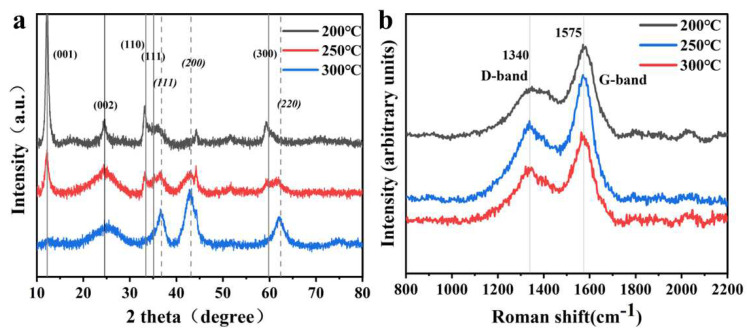
XRD spectra (**a**) and Raman patterns (**b**) of CNFs@Ni(OH)_2_/NiO-X.

**Figure 3 materials-15-08377-f003:**
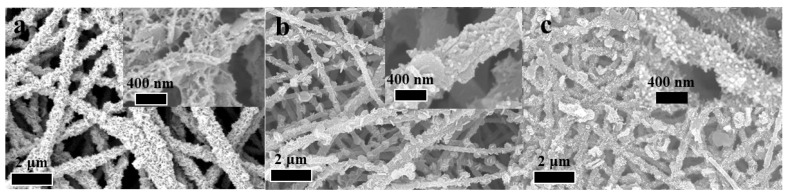
SEM pictures of CNFs@Ni(OH)_2_/NiO-X (200 (**a**), 250 (**b**) and 300 (**c**)).

**Figure 4 materials-15-08377-f004:**
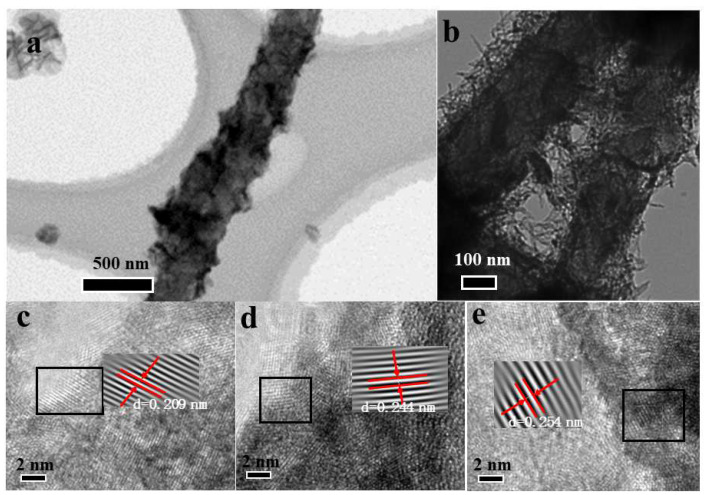
TEM pictures of CNFs@Ni(OH)_2_/NiO-250 (**a**,**b**).HRTEM pictures of CNFs@Ni(OH)_2_/NiO-250 (**c**–**e**).

**Figure 5 materials-15-08377-f005:**
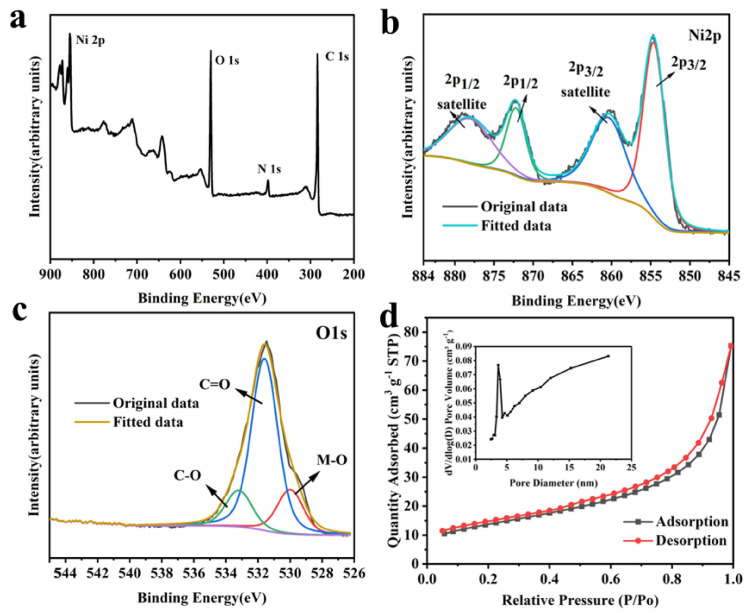
Survey (**a**), Ni2p (**b**), O1s (**c**) XPS spectra, and N_2_ adsorption-desorption isotherm as well as pore size distribution (inset) (**d**) of CNFs@Ni(OH)_2_/NiO-250.

**Figure 6 materials-15-08377-f006:**
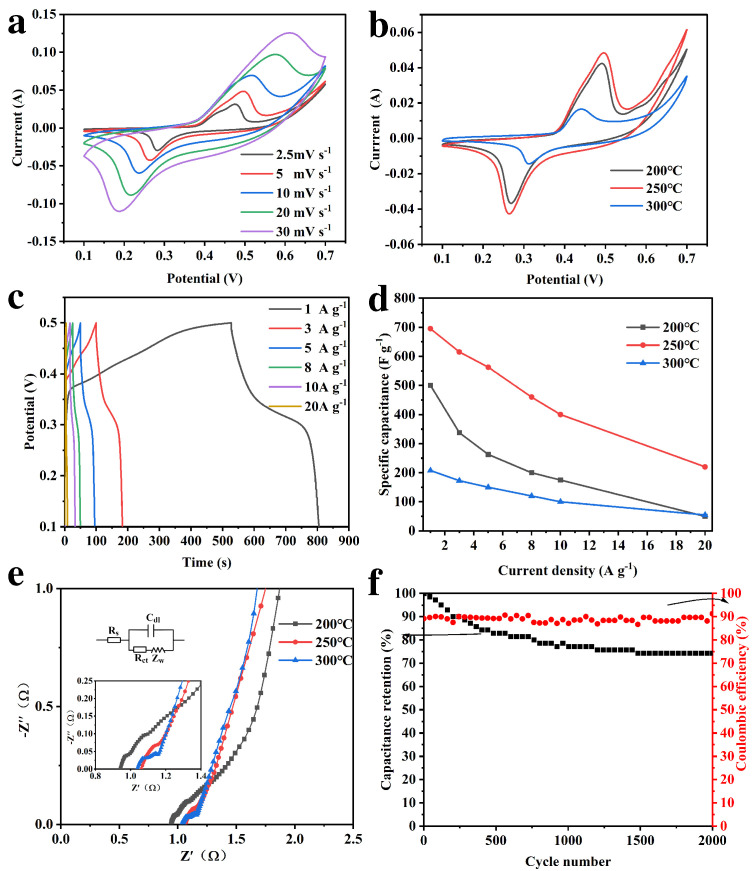
CV curves of CNFs@Ni(OH)_2_/NiO-250 at different sweep speeds (**a**). CV curves of CNFs@Ni(OH)_2_/NiO-X at 5 mV s^−1^ sweep rate (**b**). GCD curves of CNFs@Ni(OH)_2_/NiO-250 at different current densities (**c**). Specific capacitance plots of CNFs@Ni(OH)_2_/NiO-X at different current densities (**d**). Nyquist plots of CNFs@Ni(OH)_2_/NiO-X (**e**). Cycling stability plots of CNFs@Ni(OH)_2_/NiO-250 (**f**).

**Figure 7 materials-15-08377-f007:**
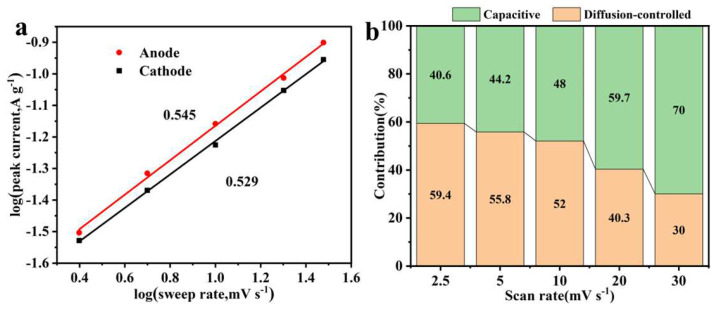
The law between log(*v*) and log(*i_p_*) of CNFs@Ni(OH)_2_/NiO-250 electrode (**a**). Bar graph of capacitance contribution at various scan rates (**b**).

**Figure 8 materials-15-08377-f008:**
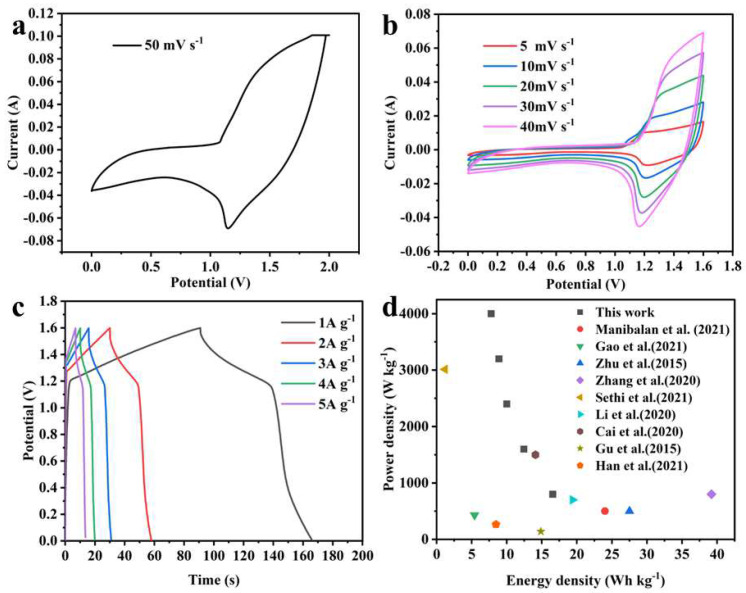
(**a**) CV curve of CNFs@ Ni(OH)_2_/NiO-250//AC ASC at 0–2 V at 50 mV s^−1^. (**b**) CV curves of ASC at diverse scan rates. (**c**) GCD curves of ASC at diverse current densities. (**d**) Ragone diagram [15,16,27,28,29,30,31,32,33].

**Table 1 materials-15-08377-t001:** Elemental contents of CNFs@Ni(OH)_2_/NiO-X.

Element	Atomic % (200)	Atomic % (250)	Atomic % (300)
Carbon	76.266	60.947	69.428
Nitrogen	1.824	1.636	2.174
Oxygen	7.013	11.671	7.662
Nickel	14.896	25.746	20.736

**Table 2 materials-15-08377-t002:** Pore properties of CNFs@Ni(OH)_2_/NiO-250.

S_A_ (m^2^ g^−1^)	V_T_ (cm^3^ g^−1^)	V_S_ (cm^3^ g^−1^)	V_I_ (cm^3^ g^−1^)	W_avg_ (nm)
49.299	0.116	0.010	0.106	7.212

*S*_A_: specific surface area. *V*_T_: total pore volume. *V*_I_: pore volume with a pore size of 2.5–50 nm. *V*_S_: pore volume with a pore size less than 2.5 nm, calculated using the formula *V*_T_–*V*_I_. *W*_avg_: desorption average pore width (nm).

## Data Availability

Not applicable.

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
