# Peer review of "Core-Shell Carbon Nanofibers@Ni(OH)2/NiO Composites for High-Performance Asymmetric Supercapacitors"

_materials, 2022, doi:10.3390/ma15238377_

Round 1

Reviewer 1 Report

1. Please include the full nomenclature with the abbreviation of chemical names.
2. What is the temperature for drying the electrode in the oven for 12 hours?
3. Please rearrange equations 3 and 4.
4. A peak near 45 degrees belongs to which material (CNF, Ni(OH)2, or NiO), and why does
its intensity increase as the temperature rises?
5. Please improve the quality of the text in Figures 4 c,d, and e. All text in d-spacing is
blurred.
6. Please check that all figure numbers, like TEM and XPS figure numbers, are the same.
7. Please check the XPS figure 4b, and add the reference for the XPS analysis.
10.1016/j.colsurfa.2022.129901
8. Why are CV and GCD potential different?
9. What is the ohmic resistance value for all three electrodes?
10. Please increase the font size of the X and Y-axis in all of the figures.
11. Please check the spelling of the cathode and anode.
12. Supporting information is missing

Reviewer 2 Report

The authors mentioned that NiOH2 and NiO are having poor stability. But why they have made both materials in their compound? They could have prepared anyone among them along with carbon materials to improve the stability.  Even after the hybrid along with CNF, they have demonstrated a stability of 74.2 % up to 1500 cycles which is significantly less % and the no of cycles is very less compared to the reports available at the present time. NiOH2 or NiO is having high theoretical specific capacity of more than 2000 F/g. However, the authors have one-third of the Cs even with CNF. The author should discuss more about the importance of this work in the introduction section.

What will happen when the cycle performance is done for more number of cycles (5000 to 10,000)

The authors have done the sample preparation with three different temperatures, which is not reflected in the abstract or at the end of the introduction section. The author may include those details also.

To emphasize this work, the author may compare their device performance with other available literature through an energy density vs power density plot.

The references can be improved with the available research reports.

Round 2

Reviewer 2 Report

The revised version may be accepted for the publication